# Atezolizumab Plus Bevacizumab in Patients with Advanced and Progressing Hepatocellular Carcinoma: Retrospective Multicenter Experience

**DOI:** 10.3390/cancers14235966

**Published:** 2022-12-02

**Authors:** Friedrich Sinner, Matthias Pinter, Bernhard Scheiner, Thomas Jens Ettrich, Niklas Sturm, Maria A. Gonzalez-Carmona, Oliver Waidmann, Fabian Finkelmeier, Vera Himmelsbach, Enrico N. De Toni, Najib Ben Khaled, Raphael Mohr, Thorben Wilhelm Fründt, Fabian Kütting, Florian van Bömmel, Sabine Lieb, Sebastian Krug, Dominik Bettinger, Michael Schultheiß, Leonie S. Jochheim, Jan Best, Christian Müller, Verena Keitel, Marino Venerito

**Affiliations:** 1Department of Gastroenterology, Hepatology and Infectious Diseases, Otto-von-Guericke University Hospital Magdeburg, 39120 Magdeburg, Germany; 2Department of Internal Medicine III, Division of Gastroenterology and Hepatology, Medical University of Vienna, A-1090 Vienna, Austria; 3Liver Cancer (HCC) Study Group Vienna, Medical University of Vienna, A-1090 Vienna, Austria; 4Ulm University Hospital, Department of Internal Medicine I, 89081 Ulm, Germany; 5Department of Internal Medicine I, University Hospital of Bonn, 53127 Bonn, Germany; 6Department of Gastroenterology, Hepatology and Endocrinology, University Hospital Frankfurt, 60590 Frankfurt, Germany; 7University Cancer Center Frankfurt, University Hospital Frankfurt, 60590 Frankfurt, Germany; 8Center for Hematology and Oncology Bethanien, 60389 Frankfurt, Germany; 9Frankfurt Cancer Institute, Goethe University Frankfurt/Main, 60590 Frankfurt, Germany; 10Department of Medicine II, University Hospital, LMU Munich, 81377 Munich, Germany; 11German Cancer Consortium (DKTK), Partner Site Munich, and German Cancer Research Center, 69120 Heidelberg, Germany; 12Department of Hepatology & Gastroenterology, Campus Virchow Klinikum and Campus Charité Mitte, Charité University Medicine Berlin, 10117 Berlin, Germany; 13Department of Internal Medicine, Gastroenterology & Hepatology, University Medical Center Hamburg-Eppendorf, 20246 Hamburg, Germany; 14Clinic for Gastroenterology and Hepatology, Faculty of Medicine and University Hospital Cologne, University of Cologne, 50937 Cologne, Germany; 15Department of Medicine II, Division of Hepatology, Leipzig University Medical Center, 04103 Leipzig, Germany; 16Department of Internal Medicine I, Martin-Luther-University Halle-Wittenberg, 06120 Halle, Germany; 17Department of Medicine II, Medical Center University of Freiburg, Faculty of Medicine, University of Freiburg, 79106 Freiburg, Germany; 18Department of Gastroenterology and Hepatology, University Hospital Essen, 45147 Essen, Germany; 19Department of Internal Medicine, University Hospital Knappschaftskrankenhaus Bochum, Ruhr University Bochum, 44892 Bochum, Germany

**Keywords:** hepatocellular carcinoma, liver cancer, atezolizumab, bevacizumab, immune checkpoint inhibition, immunotherapy

## Abstract

**Simple Summary:**

Hepatocellular cancer is the most common type of primary liver cancer. It is the third leading cause of cancer-related deaths worldwide and its incidence is increasing: >1 million new cases per year expected by 2025. Despite advances in treatment in recent years, diagnosis is associated with poor overall survival. Treatment of hepatocellular cancer depends on the patient’s general health and fitness, how well the liver is working, the number and size of tumors in the liver, and whether or not the tumor has spread to other neighboring or distant parts of the body. The combination of atezolizumab plus bevacizumab, two intravenously administered antibodies, is the preferred first-line treatment for patients with advanced hepatocellular cancer that has spread from the liver to other neighboring or distant parts of the body. This study investigated how long patients whose hepatocellular cancer continues to grow (progress) despite one or more prior tumor therapies live when they receive atezolizumab plus bevacizumab. These patients, treated with atezolizumab plus bevacizumab at various hospitals in Germany and Austria, lived about 16 months, which is about 5–8 months longer than patients receiving approved drugs. The safety profile was consistent with previous reports.

**Abstract:**

Atezolizumab plus bevacizumab is the standard of care for first-line systemic therapy for advanced hepatocellular carcinoma (aHCC). Data on the efficacy and safety of atezolizumab plus bevacizumab in patients with aHCC who have received prior systemic therapy are not available. **Methods:** Patients with aHCC who received atezolizumab plus bevacizumab after at least one systemic treatment between December 2018 and March 2022 were retrospectively identified in 13 centers in Germany and Austria. Patient characteristics, tumor response rates, progression-free survival (PFS), overall survival (OS), and adverse events (AE) were analyzed. **Results:** A total of 50 patients were identified; 41 (82%) were male. The median age at initiation of treatment with atezolizumab plus bevacizumab was 65 years, 41 (82%) patients had cirrhosis, 30 (73%) Child A, 9 (22%) B, and 2 (5%) C. A total of 34 patients (68%) received atezolizumab plus bevacizumab in the second-line setting and 16 (32%) in later lines. The best radiologic tumor responses were complete remission (2%), partial remission (30%), stable disease (36%), and progressive disease (18%), resulting in an objective response rate of 32% and a disease control rate of 68%. Median OS was 16.0 months (95% confidence interval 5.6–26.4 months), and median PFS was 7.1 months (95% confidence interval 4.4–9.8 months). AE grades 3–4 were observed in seven (14%) and resulted in death in three patients (6%). There were five (10%) bleeding events with a grade ≥ 3, including one (2%) with a fatal outcome. **Conclusions:** Atezolizumab plus bevacizumab is effective in patients with aHCC who did not have access to this option as first-line therapy. The safety profile was consistent with previous reports.

## 1. Introduction

Hepatocellular carcinoma (HCC) accounts for roughly 80% of primary liver cancer, which is the third most leading cause of cancer-related deaths worldwide and affects more than 800,000 patients annually [1,2]. Patients with early stage HCC are eligible for treatments with curative intent, including local ablative procedures, surgical resection or liver transplantation, whereas systemic treatment is the therapy of choice for locally advanced or metastatic disease [3,4].

Since its approval in 2007, the tyrosine-kinase inhibitor (TKI) sorafenib has been the only available treatment for advanced-stage HCC (aHCC) for more than a decade [5,6]. However, since 2018, several new agents have been shown to be effective and have been approved for the treatment of aHCC. The TKI lenvatinib was noninferior to sorafenib in the phase III REFLECT trial, and represents an alternative first-line therapy [7], while three placebo-controlled phase III trials in eligible candidates with aHCC showed a survival benefit of approximately 3 months with either the TKI cabozantinib, or the TKI regorafenib for patients who had previously tolerated sorafenib, or the anti-VEGF2 monoclonal antibody (mAb) ramucirumab for patients with alpha-fetoprotein (AFP) ≥ 400 ng/ml [8,9,10].

Despite the immense breakthrough in cancer therapy with the discovery of monoclonal antibodies targeting immune checkpoints, no consistent survival benefit was achieved for patients with aHCC by monotherapy with the anti-programmed cell death 1 (PD-1) mAb nivolumab and pembrolizumab in the first- and second-line setting, respectively [11,12]. The intuition that combined inhibition of VEGF and PD-L1 may be the key to effective treatment of aHCC represents a paradigm shift in the treatment landscape of this tumor. In November 2019, the phase III IMbrave150 trial showed a survival benefit of about six months over sorafenib for aHCC patients treated with the anti-PD-L1 mAb atezolizumab in combination with the anti-VEGF mAb bevacizumab administered in first-line systemic treatment [13,14]. This trial led to the approval of atezolizumab plus bevacizumab for patients with aHCC who have not received prior systemic therapy [15,16]. Unfortunately, over the course of this revolution, patients with aHCC who had already received first-line therapy with sorafenib or lenvatinib were excluded from the most effective treatment available. Indeed, there are no reliable data on atezolizumab plus bevacizumab after first-line therapy.

Accordingly, apart from the ASCO guideline, which states that atezolizumab plus bevacizumab may be considered following first-line sorafenib or lenvatinib therapy “where patients did not have access to this option as first-line therapy,” [17] this recommendation is not supported by any other HCC guideline [3,18,19,20]. However, evidence from clinical practice and the peculiar mechanism of action of immune checkpoint inhibitors suggest that the use of atezolizumab plus bevacizumab is a rational option in patients with aHCC who progress after previous systemic therapies when other established therapeutic options are unavailable or contraindicated.

We present real-world data on the efficacy and safety of atezolizumab plus bevacizumab in patients with aHCC who have received at least one prior systemic therapy from 13 centers in Germany and Austria.

## 2. Material and Methods

### 2.1. Study Design and Selection of Patients

This is a post-approval retrospective study conducted to obtain information on the efficacy and safety of atezolizumab plus bevacizumab in patients with aHCC who have received at least one prior systemic therapy. Patients with histologically or radiologically confirmed aHCC who received at least one cycle of atezolizumab plus bevacizumab in the second- or further-lines between December 2018 and March 2022 were retrospectively identified at 13 centers in Germany and Austria. Consistent with a real-world dataset, no further inclusion and exclusion criteria were defined.

Demographic data, underlying liver disease, laboratory results, tumor-specific characteristics such as date of diagnosis, tumor stage at the time of diagnosis according to portal invasion, extrahepatic spread (EHS), and previous treatments, were retrospectively assessed. Liver function was classified using the Child–Pugh (CP) score, which is based on the presence of liver cirrhosis. Patients with HCC in cirrhosis were staged according to the Barcelona Clinical Liver Cancer (BCLC) classification. Patients were also assessed by the albumin–bilirubin (ALBI) score.

Adverse events (AEs) were graded according to the common terminology criteria for adverse events (CTCAE) version 5.0.

Response to treatment was assessed by the best radiological response under treatment by computed tomography or magnetic resonance imaging. Response was graded as complete and partial response (CR and PR), stable disease (SD), and progressive disease (PD) by local review according to Response Evaluation Criteria in Solid Tumors (RECIST) 1.1. Objective response rate (ORR) was defined as the proportion of patients achieving CR or PR. Disease control rate (DCR) was defined as the proportion of patients achieving a CR, PR, or SD as the best radiologic response. In addition to radiological response, AFP levels at baseline and during the treatment were documented. In addition to tumor progression, other reasons for treatment discontinuation such as worsening liver function and other AEs were analyzed. Patients were followed until death or data cut-off (31 March 2022). Patients whose last documented visit occurred more than 3 months before the data cut-off were considered lost to follow-up.

Atezolizumab in combination with bevacizumab has been approved by the European Medicines Agency (EMA), and the United States Food and Drug Administration (FDA) since 2020 for the treatment of patients with unresectable or advanced HCC who have not received prior systemic therapy. The recommended atezolizumab dose for HCC is 1200 mg i.v. followed by bevacizumab 15 mg/kg i.v. on the same day, every 3 weeks.

### 2.2. Statistical Analysis

Baseline characteristics were summarized using descriptive statistics with numbers, percentages, and median with ranges. Data are presented as medians and full ranges for continuous variables, and frequencies and percentages for categorical variables. Median duration of treatment was defined as time from the date of the first administration until the date of the last documented administration.

Overall survival (OS) was defined as the period of survival from the day treatment was initiated until the day death occurred. The date of last contact or data cutoff was used to censor patients who were still alive. Patients still receiving atezolizumab plus bevacizumab were censored at the time of data cut-off.

Progression-free survival (PFS) was defined as the time from the date of first administration of atezolizumab plus bevacizumab to radiologically confirmed disease progression or death, whichever occurred first. Patients who were alive and did not have radiologically confirmed progression were censored at the time of last contact or data cut-off. Patients who had at least one imaging follow-up were evaluable for radiologic response. Patients lost to follow-up without prior radiologic progression could not be evaluated.

OS and PFS were calculated using Kaplan–Meier survival analysis. Differences between groups were analyzed using the log rank test and expressed as median with the corresponding 95% confidence intervals (95% CI). Univariate and multivariate Cox regression models with stepwise likelihood ratio (forward selection) were used to analyze independent prognostic parameters. Sex, age, APF level, CP and BCLC stage, ALBI grade, presence of cirrhosis, EHS, portal vein thrombosis or infiltration, and prior ICI treatment were included in the model. Results were presented as hazard ratios (HR) with the corresponding 95% CI. Variables with statistical significance (*p* < 0.05) were included in the multivariate analysis. All tests were two-sided, and a *p* value < 0.05 was considered statistically significant. Statistical analysis was performed using SPSS (version 28.0, IBM, New York, NY, USA).

## 3. Results

### 3.1. Patients

A total of 50 patients from 13 centers (12 German centers and 1 Austrian center), were included. The main baseline characteristics are shown in Table 1. Individual patients had started treatment with atezolizumab plus bevacizumab in the time from 17 December 2018 to 26 June 2021. Data cut-off for the analysis was 31 March 2022. Forty-one patients (82%) were male and the median age at initiation of treatment with atezolizumab plus bevacizumab was 65 years with a range of 50–80 years. Most patients (*n* = 41, 82%) had cirrhosis, with CP stage A, B, and C in 30 (73%), 9 (22%), and 2 (5%), respectively. The majority of patients were staged as C according to BCLC criteria (23 of 41 patients with liver cirrhosis, 56%). The most common risk factors for HCC were alcohol consumption in 13 (25%), nonalcoholic fatty liver disease (NAFLD) in 12 (23%), hepatitis C virus (HCV) infection in 11 (21%), and hepatitis B virus (HBV) infection in 6 (12%) patients. When categorized by ALBI grade, 19 patients (38%) had grade A, 28 (56%) had B, and 3 (6%) had C.

A total of 34 patients (68%) received atezolizumab plus bevacizumab in the second-line setting and 16 (32%) in further lines. First-line treatment was a TKI in 42/50 patients. The median observation time was 10.1 months (range 0.1–25.3 months). At the last follow-up, 23 patients had died, 22 were alive, and 5 were no longer on follow-up. Additional baseline characteristics are shown in Table 1.

Reimbursement for off-label therapies was requested individually from the responsible health insurance company prior to the start of treatment and reimbursed on a case-by-case basis. Apart from the high remission rates reported in the Imbrave150 trial, off-label therapy with atezolizumab plus bevacizumab was often motivated by adverse events observed during prior TKI therapy, contraindications to TKIs (40%), or low AFP precluding therapy with ramucirumab (36%).

### 3.2. Treatment Response

Median overall survival (mOS) from the start of first systemic treatment in the overall cohort was 31.6 months (95% CI 9.0–51.2). The median duration of treatment with atezolizumab plus bevacizumab was 7.0 months and ranged from 0.1 to 22.3 months. Median progression-free survival (mPFS) was 7.1 months (95% CI 4.4–9.8 months) and mOS after initiation of treatment with atezolizumab plus bevacizumab was 16.0 months (95% CI 5.6–26.4) months (Figure 1 and Figure 2). Patients receiving atezolizumab plus bevacizumab in the second-line had a mOS of 16 months (95% CI 1.5–30.5), compared to 17.7 months (95% CI 3.8–31.6) in patients receiving this combination in later lines (HR 0.85, 95% CI 0.35–2.1, *p* = 0.722, Figure 3). The mPFS was 6.3 months (95% CI 3.9–8.7 months) for patients who received atezolizumab plus bevacizumab in the second-line and 8.5 months (95% CI 6.5–10.5 months) in later lines (HR 0.69, 95% CI 0.32–1.45, *p* = 0.322).

Patients with CP-A achieved a mOS of 16.0 months (95% CI, 5.6–26.4), whereas it was 6.4 months (95% CI, 0.0–13.1) in patients with CP-B (*p* = 0.200). Patients with CP-A achieved a mPFS of 8.7 months (95% CI, 4.5–12.9), while it was 3.4 months (95% CI, 1.4–5.4) in patients with CP-B (*p* = 0.008).

The mOS was not reached in patients with ALBI grade 1, while it was 6.4 months (95% CI, 4.5–8.3) in patients with ALBI grade 2 and 6.3 months (95% CI, 0.2–12.4) in patients with ALBI grade 3.

Compared to patients with preserved liver function according to the ALBI grade (ALBI grade 1), survival of patients with impaired liver function was significantly shorter (ALBI 1 vs. ALBI 2: HR 3.4, 95% CI 1.14–10.30, *p* = 0.029; ALBI 1 vs. ALBI 2: HR 6.6, 95% CI: 1.46–29.89; *p* = 0.014). The mPFS for patients with ALBI grade 1, 2, and 3 was 7.6 months (95% CI, 5.3–9.9), 6.7 months (95% CI, 3.8–9.6), and 3.8 months (95% CI, 1.2–6.4), respectively, with no statistical difference among groups.

Radiologic response was assessed in 43 patients (86%) according to RECIST criteria v1.1. Of these patients, 1 (2%) achieved a complete response (CR), 15 (30%) achieved a partial response (PR), and 18 (36%) had a stable disease (SD), while 9 (18%) had progressive disease (PD). The overall response rate (ORR) was 32%, and the disease control rate (DCR) was 68% (Table 2).

Response rates were similar in patients who received atezolizumab plus bevacizumab after one, two, or more prior lines of systemic treatment (Table 2).

OS was not significantly different between patients with AFP levels ≥ 400 ng/mL compared to patients with low AFP levels (*p* = 0.744). The median OS was 17.7 months (95% CI 2.0–33.4 months) in patients with AFP levels > 400 ng/mL; compared to patients with lower AFP levels, mOS was 16.0 months (95% CI 3.6–28.4 months). Patients who had previously received treatment with a checkpoint inhibitor had similar mOS compared to patients without ICI therapy prior to the initiation of treatment with atezolizumab plus bevacizumab (*p* = 0.81).

### 3.3. Safety

All patients included in the present analysis received at least one dose of atezolizumab plus bevacizumab and were monitored for the development of treatment-related adverse events (TRAE). At the time of data cut-off, 38 patients (76%) had discontinued treatment with atezolizumab plus bevacizumab or had died. In four cases (8%), the bevacizumab medication was paused due to TRAE or planned surgery, in three cases (6%), bevacizumab was not continued. In 25 patients (50%) at least one TRAE was reported, whereas the total number of observed TRAEs was 36. Events of grade 3–4 were observed in seven (14%) cases and led to the death of three patients (6%). The most common TRAEs were bleeding events in seven (14%), rash/exanthem in six (12%), and fatigue in four (8%) patients. There were five (10%) bleeding events of grade ≥ 3, one of them (2%) with fatal outcome was reported. The incidence of each TRAE is shown in Table 3.

### 3.4. Prognostic Markers Associated with Survival

Apart from a significant correlation of OS with the ALBI score (HR 3.42, 95% CI 1.14–10.30 for ALBI grade 1 vs. 2 and HR 6.6, 95% CI 1.46–29.89 for ALBI grade 1 vs. 3), no other prognostic markers were identified. In the multivariate Cox regression model, ALBI score was independently associated with OS (HR 0.05, 95% CI 0.004–0.52 for ALBI grade 1 vs. 2 and HR 0.13, 95% CI 0.013–1.23 for ALBI grade 1 vs. 3, respectively).

## 4. Discussion

To our knowledge, this is the largest cohort in the literature of patients with aHCC receiving atezolizumab plus bevacizumab after prior systemic therapy. We have shown that atezolizumab plus bevacizumab resulted in higher ORR, PFS, and OS than expected with standard treatment [8,9,10] in patients who had disease progression after at least one systemic treatment for aHCC, regardless of serum AFP level. In addition, both the efficacy and safety profiles of atezolizumab plus bevacizumab in our cohort were similar to those in patients treated in the IMbrave150 trial, which did not allow prior systemic treatment. Thus, the results of this multicenter retrospective study support the clinical use of atezolizumab plus bevacizumab in patients who did not have access to this option as a first-line therapy in the pre-approval period, who had contraindications to established therapies after first-line TKI-based therapy, or for whom no other approved treatment was available.

Because 32% of patients received atezolizumab plus bevacizumab as a third-line treatment or later, the favorable survival of these patients may be due in part to less aggressive tumor biology rather than treatment effect. However, an objective response was seen in both this subset of patients who had received at least two prior systemic therapies and in the larger proportion of patients (68%) who received atezolizumab plus bevacizumab as a second-line therapy, strongly supporting the evidence for the intrinsic efficacy of atezolizumab plus bevacizumab.

Approximately 20% of patients in our cohort had impaired liver function (CP B). However, the efficacy and safety of atezolizumab plus bevacizumab was independent of the CP stage in our cohort. This is consistent with recently published data on the use of atezolizumab plus bevacizumab [21] in first-line treatment. Nevertheless, multivariate Cox regression identified liver function (ALBI score) as the strongest prognostic marker in our cohort, underscoring the known effect of liver function in determining prognosis in these patients [22,23].

Obvious limitations of our study are its retrospective nature and the lack of central radiological assessment at predefined intervals. On the other hand, a positive effect of atezolizumab plus bevacizumab was observed, although these patients had unfavorable baseline characteristics that would have prevented their inclusion in clinical trials: these include the fact that many patients who received atezolizumab plus bevacizumab had contraindications to TKI-based treatment, poor performance status (12% of patients had a ECOG PS of 2) and/or poorer liver function (27% had CP B or C). Despite these limitations, atezolizumab plus bevacizumab proved safe and demonstrated clinical efficacy after one or more prior lines of treatment, exceeding the ORR, PFS, and OS reported in contemporary studies.

A number of different immune checkpoint inhibition (ICI)-based treatment strategies have been evaluated in global phase III clinical trials. The ICI monotherapy with durvalumab was shown to be non-inferior to the TKI sorafenib with improved safety profile [24]. However, among combination therapies, only the ICI combination with durvalumab and tremelimumab provided a survival benefit over sorafenib [24], whereas both the COSMIC-312 and LEAP-002 trials, which investigated the superiority of a combination TKI and ICI compared with TKI monotherapy, provided negative results with respect to mOS [25,26]. Of note, in the COSMIC-312 trial, the combination of the TKI cabozantinib with the ICI atezolizumab was superior to a TKI monotherapy in terms of PFS, whereas the significantly higher percentage of patients who received subsequent therapy with VEGF(R)-targeted antibody or ICI in the standard arm may have contributed to preventing a mOS benefit in the experimental arm. These results are consistent with our retrospective analysis and indicate that the bar has been raised for future first-line trials in aHCC. While future studies on the optimal treatment sequence for patients with advanced HCC are still pending, our results support current ASCO recommendations and should be considered in other international guidelines.

## Figures and Tables

**Figure 1 cancers-14-05966-f001:**
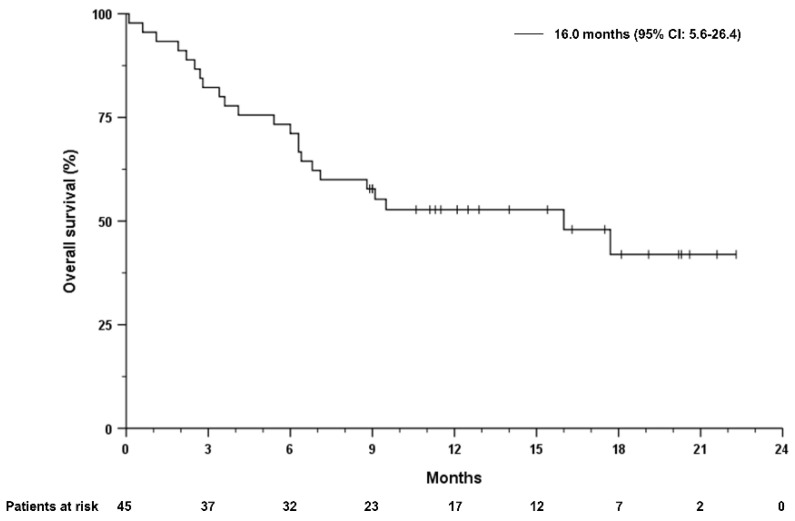
Overall survival of the overall study cohort, independent of the therapy line. Overall survival was defined by the date of treatment start to death from any cause. Tick marks indicate censored data.

**Figure 2 cancers-14-05966-f002:**
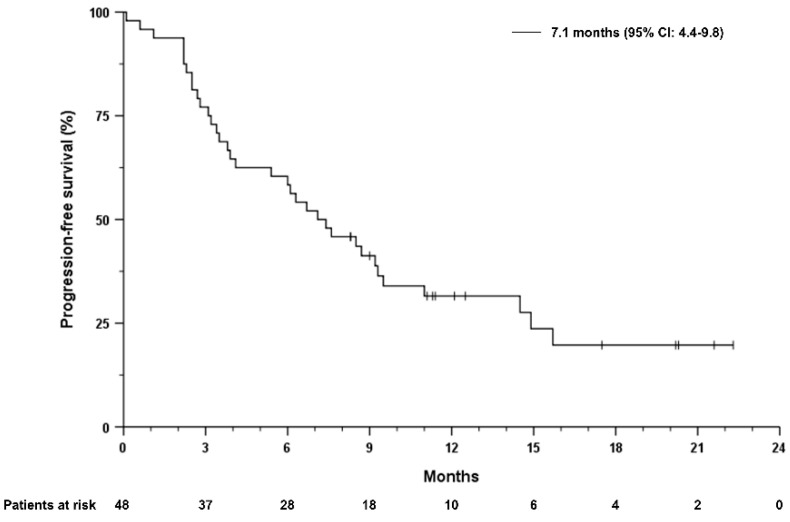
Progression-free survival of the overall study cohort, independent of the therapy line. Progression-free survival as the time from treatment start to radiographic progression or death from any cause. Tick marks indicate censored data.

**Figure 3 cancers-14-05966-f003:**
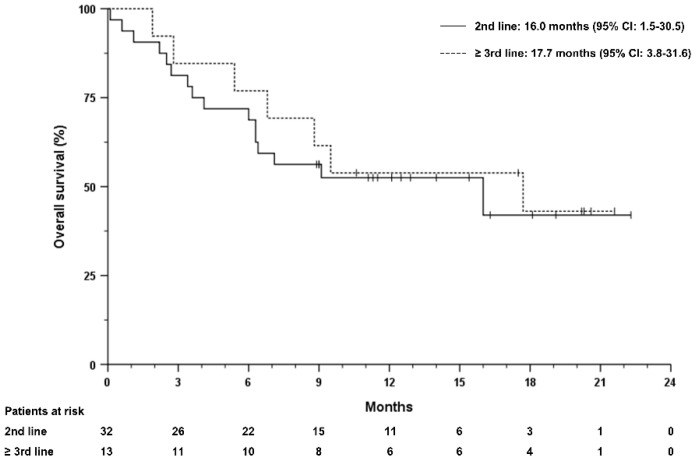
Overall survival stratified according to the number of previous therapy lines. Overall survival was defined by the date of treatment initiation until death from any cause. Tick marks indicate censored data.

**Table 1 cancers-14-05966-t001:** Baseline characteristics.

Parameters	Patients, *n* = 50 (%) *
Median age, years (range)	65 (50–80)
Gender, male/female	41 (82)/9 (18)
ECOG, 0/1/2	27 (54)/17 (34)/6 (12)
Liver cirrhosis present	41 (82)
Child-Pugh class, A/B/C	30 (73)/9 (22)/2 (5)
BCLC stage, B/C/D	7 (17)/23 (56)/11 (27)
ALBI score, grade 1/2/3	19 (38)/28 (56)/3 (6)
Risk factors for HCC: HCV/HBV/alcohol/NAFLD	11 (21)/6 (12)/13 (25)/12 (23)
/other **/none	/5 (10)/5 (10)
AFP ≥ 400 ng/mL	32 (64)
EHS	27 (54)
Portal invasion	17 (34)
EHS and/or portal invasion	37 (74)
Prior surgical treatment	20 (40)
Prior loco-regional treatment	35 (70)
Prior loco-regional and/or surgical treatment	40 (80)
Systemic treatment in first-line, lenvatinib/sorafenib/other ***	26 (52)/16 (32)/11 (22)
Received atezolizumab plus bevacizumab in line, 2/3/4/5	34 (68)/9 (18)/6 (12)/1 (2)
Vital status at last follow-up, dead/alive/unknown	23 (46), 22 (44), 5 (10)
Median observation period, months (range)	10.1 (0.1–25.3)

Abbreviations and notes: AFP, αlpha-fetoprotein; ALBI, albumin-bilirubin; BCLC, Barcelona Clinical Liver Cancer; ECOG, Eastern Cooperative Oncology Group; EHS, extrahepatic spread; HBV, hepatitis B virus; HCC, hepatocellular carcinoma; HCV, hepatitis C virus; NAFLD, nonalcoholic fatty liver disease; * Due to rounding, the percentage may differ from the total. ** hemochromatosis, *n* = 3; Wilson’s disease, *n* = 1; beta-catenin adenoma, *n* = 1. The sum is higher than 50 as 2 patients had 2 risk factors. *** cabozantinib *n* = 2; nivolumab *n* = 2, pembrolizumab plus regorafenib *n* = 1; pembrolizumab plus envatinib *n* = 1; spartalizumab plus sorafenib *n* = 1; tislelizumab *n* = 1; regorafenib *n* = 1. The sum is higher than 50 as 3 patients received a combination of systemic therapy in first line.

**Table 2 cancers-14-05966-t002:** Response rates.

Best Documented Response	Patients, *n* = 50 (%)	2nd Line, *n* = 34 (%)	≥3rd Line, *n* = 16 (%)
Complete response (CR)	1 (2)	1 (3)	0 (0)
Partial response (PR)	15 (30)	10 (29)	5 (31)
Stable disease (SD)	18 (36)	12 (35)	6 (38)
Progressive disease	9 (18)	5 (15)	4 (25)
Not evaluable	7 (14)	6 (18)	1 (6)
Objective response rate (ORR)	16 (32)	11 (32)	5 (31)
Disease control rate (DCR)	34 (68)	23 (68)	11 (69)

Notes: Radiological response was available for 43 patients (86% of the efficacy population). The percentages may differ from the total due to rounding.

**Table 3 cancers-14-05966-t003:** Treatment-related adverse events (TRAE) during the treatment in the safety population.

TRAE	Any Grade, *n* (%)	Grade 1–2, *n* (%)	Grade 3–4, *n* (%)	Death, *n* (%)
Rash/exanthema	6 (12)	6 (12)	0	0
Esophageal variceal bleeding	4 (8)	0	3 (6)	1 (2)
Fatigue	4 (8)	4 (8)	0	0
Thyroid toxicity	3 (6)	3 (6)	0	0
Hepatotoxicity/hepatitis	2 (4)	0	1 (2)	1 (2)
Epistaxis	2 (4)	2 (4)	0	0
Hyponatremia	2 (4)	2 (4)	0	0
Hypertension	2 (4)	2 (4)	0	0
Pruritus	2 (4)	2 (4)	0	0
Hypoglycemia	1 (2)	0	0	1 (2)
Retroperitoneal bleeding	1 (2)	0	1 (2)	0
Hyperglycemia	1 (2)	0	1 (2)	0
Worsening asthma	1 (2)	0	1 (2)	0
Hepatic encephalopathy	1 (2)	1 (2)	0	0
Dyspnea	1 (2)	1 (2)	0	0
Dysphonia	1 (2)	1 (2)	0	0
Infusion reaction	1 (2)	1 (2)	0	0
Appetite loss	1 (2)	1 (2)	0	0

## Data Availability

All data generated or analyzed during this study are included in this article. Further inquiries can be directed to the corresponding author.

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
