# Peer review of "Atezolizumab Plus Bevacizumab in Patients with Advanced and Progressing Hepatocellular Carcinoma: Retrospective Multicenter Experience"

_cancers, 2022, doi:10.3390/cancers14235966_

Round 1
Reviewer 1 Report
1. must provide clinical characteristics for HCC cohort.
2. Include the virus infection but not illustrate the viral treatment response.
3. Occurrence HCC time needs to calibrate for the statistic.
Author Response
- must provide clinical characteristics for HCC cohort.
- We received an Email from the editorial office of Cancers on the 21 of October noting that the tables of our manuscript were not uploaded. This mistake was fixed in the very next day. We noted that reviewer 1 wrote his comments on the 17. October 2022, thus without seeing the tables. The clinical characteristics are available in the current version of the manuscript.
- Include the virus infection but not illustrate the viral treatment response.
- In the current version of the manuscript, a table with the etiology of HCC is provided.
- Treatment response to the subgroup of patients with viral etiology was not in the focus of the present manuscript.
- Occurrence HCC time needs to calibrate for the statistic.
- The overall survival has been analysed from the start of Atezo/Bev as well as from the first systemic therapy. Previous non-systemic therapies including surgical and locoregional therapies are shown in the baseline characteristic table as well.
- Extensive editing of English language and style required
- A native English-speaking colleague has checked the manuscript.
Reviewer 2 Report
The authors provide efficacy and safety real-world data for atezolizumab plus bevacizumab in patients with advanced HCC who received at least one previous systemic therapy, identified across 13 centers in Germany and Austria. They concluded that atezolizumab plus bevacizumab proved safe and showed clinical efficacy after one or more prior lines of treatment, exceeding the ORR, PFS, and OS reported in contemporary studies, respectively.
Unfortunately, it is a retrospective study of a small number of cases, and there are prior papers with numerous cases, so it lacks originality (PMID: 34114752).
Author Response
Unfortunately, it is a retrospective study of a small number of cases, and there are prior papers with numerous cases, so it lacks originality (PMID: 34114752).
--> The retrospective nature of the study is an obvious limitation. The number of cases, though small, is the highest achievable using an off-label therapy in Europe, where the therapy indications are strictly monitored. In the paper cited by Reviewer2 on early experience with Atezo/Bev from a Japanese cohort, survival data are not shown.
Reviewer 3 Report
The article entitled "Atezolizumab plus bevacizumab in patients with advanced and progressing hepatocellular carcinoma: retrospective multicenter experience” by Sinner and cols. is a first and necessary manuscript considering the lack of data regarding the management of patients with advanced HCC previously treated with TKIs or with progressing HCC.
The study, although multicenter, has a limited (small) number of patients, which is a drawback considering that some of the observed outcomes refer to a single patient (2%), or a very small number of patients. But even so, in general, the patients treated behave in a similar way to that observed in patients with atezolizumab plus bevacizumab in the first line, which gives it credibility. Another drawback is its retrospective nature, but still it is interesting to publish.
In any case, the limitations of the abstract study should be further underlined, in order to encourage other groups/hospitals to replicate these data in patients with similar characteristics, and thus be able to offer them a reliable treatment alternative.
Given the nature of the study, there are no major concerns other than stressing its limited nature.
Author Response
- Thank you for your thoughts. We completely agree with Reviewer 3. The number of cases, though small, is the highest achievable using an off-label therapy in Europe, where the therapy indications are strictly monitored.
- We stress the retrospective nature of the study both in the title and in the methods section of the abstract.
Round 2
Reviewer 1 Report
The authors provide sufficient research results according to the previous edition. The article reaches publication in the "Cancers.